



# Neutral air turbulence in the mesosphere and associated polar mesospheric summer echoes (PMSEs)

Alireza Mahmoudian[1], Mike J. Kosch[2,3,4], Wayne A. Scales[5], Michael T. Rietveld[6], and Henry Pinedo[7]

[1]Institute of Geophysics, University of Tehran, Iran.
[2]Department of Physics, Lancaster University, Lancaster, UK.
[3]South African National Space Agency (SANSA), Hermanus, South Africa
[4]Dept. of Physics and Astronomy, University of the Western Cape, Bellville, South Africa
[5]Bradley Department of Electrical and Computer Engineering, Virginia Tech.
[6]EISCAT Scientific Association, Ramfjordmoen, Norway
[7]Department of Physics and Technology, University of Tromsø, Tromsø, Norway

**Correspondence:** Alireza Mahmoudian (a.mahmoudian@ut.ac.ir)

**Abstract.** This paper presents the first simultaneous four radar frequency observations of the PMSE region under varying neutral air turbulence conditions. Radar frequencies of 8, 56, 224, and 930 MHz are used in this study. Three days of experimental observations associated with EISCAT are presented. Numerical simulations of mesospheric dusty/ice plasma associated with the observed radar frequencies are presented. The effect of neutral air turbulence on the generation and strength of plasma
density perturbations associated with PMSE using four radar frequencies and in the presence of various dust parameters is investigated. Using the model it is shown that the well-known neutral air turbulence in presence of heavy dust particles, so-called fossil turbulence, can largely explain the observed radar cross-section at four radar frequencies. The effect of initial turbulence amplitude along with dust charging and diffusion in the presence of various dust parameters is investigated using the computational model. Specifically, the response of diffusion to charging time scales, plasma density fluctuation amplitude
to the background dusty plasma parameters are discussed. Several key parameters in dusty plasma responsible for the PMSE observations are determined.

## 1 Introduction

So-called Polar Mesospheric Summer Echoes PMSEs are very strong coherent radar echoes produced by electron density
fluctuations at half the radar wavelength (Bragg scatter condition) in the summer polar mesosphere (80-90 km). While polar mesospheric echoes have been observed in the absence of neutral air turbulence in some cases (Lubken et al., 1993; 2002), theoretical studies by Hill et al. (1999) and Rapp and Lubken (2004) have shown that coupling of the neutral air turbulence with the dusty plasma is the main driving source for PMSE and the associated electron density fluctuations in the mesopause region. Without including dust particles, the small-scale electron density fluctuations produced through coupling of neutral air





turbulence with electron density fluctuations, diffuse out very fast due to high viscosity effect. Schmidt number $Sc$ is defined

as a ratio of viscosity $\nu$ to electron diffusion coefficient $D$ $(Sc = \nu/D)$ (Lubken et al., 1998). Schmidt number is unity if dust

particles are excluded. In other words, the spectrum of electron density fluctuation will have the same cut-off as the neutral air

density fluctuations when no dust particles exist. The spectrum of velocity fluctuations in a turbulent medium is scaled with

of $k^{-5/3}$ where $k$ is the wavenumber. It has been shown that presence of aerosol particles in the mesosphere can increase the

Schmidt number to values much greater than unity and extend the viscous cut-off (Cho et al., 1992; Cho and Kelley, 1993;

Cho and Rottger, 1997). In addition to the role of charged aerosols in reducing the diffusion timescale of electron density

fluctuation, charged aerosol may also result in radar scatter. The so-called dressed aerosol scatter may increase radar scatter

above the incoherent scatter and is not dependent on radar wavelength. This theory can explain the observed PMSE using

UHF radars. Electron density fluctuations observed in recent in-situ measurements using rocket probes have shown a good

agreement with the theory of neutral air turbulence coupling with charged species (Rapp and Lubken, 2004; Lie-Svendsen et

al., 2003). Fluctuations in dusty plasma may also be generated by "fossil turbulence" when neutral air turbulence is absent

(Cho and Rottger, 1997; Rapp and Lubken, 2004). While velocity field is the driving source for active turbulence, electric

field is the generation source for fossil turbulence and fluctuation in plasma and dust densities. It can be shown that electron

and ion density fluctuations produced by fossil turbulence are out of phase. The spatial scale of fossil turbulence is typically

described by $\omega \ll kc_{ns}$ where $c_{ns}$ is the neutral sound speed (Mahmoudian et al., 2017b). While there are several theories

proposed to justify the electron diffusion survived long after the source (neutral air turbulence) has stopped is still an open

question in community. La Hoz et al. (2006) used the theoretical model of Hill (1978) which included multipolar diffusion

to determine the associated Schmidt numbers. The results required large number of electrons of the order of 10 on the dust

particles for enhanced scattering beyond Batchelor scale (Batchelor, 1959). Such a high charge number density may result

in electron density bite-out and is far from the range of dust radii observed using sounding rockets in the polar mesosphere

(Robertson et al., 2009). The present study is the first attempt to provide a detail study on dusty plasma parameterization of

radar echoes in the presence of mesospheric neutral air turbulence and dust particles.

The computational modeling of mesospheric plasma mixed with ice/dust particles which are capable of attracting free ions

and electrons has been studied by Lie-Svenson et al. (2003) and Scales (2004). These first studies developed the concept

of charging of the mesospheric electrons onto irregular dust density leave electron density "fingerprints" in the form of the

electron irregularities that produce PMSE. More simulation work has been done to investigate the temporal evolution of electron

irregularities in response to radio wave heating (Mahmoudian et al., 2011; Senior et al., 2014; Mahmoudian et al., 2017a). Very

recent work by Mahmoudian et al. (2018) is dedicated to the time evolution of PMSE during enhanced electron density in the

mesosphere produced by electron precipitation events. This work demonstrated that the ratio of electron density fluctuation

amplitude $\delta n_e$ to the plasma density $(n_e)$ plays a critical role in the appearance/disappearance of the layer. The simulation

results also revealed that the existence of PMSE is mainly determined by dust radius and dust density.

The coupling of the neutral air turbulence with mesospheric dusty plasma as a generation source of fluctuations in plasma

and dust densities as well as electric field is investigated by Mahmoudian et al. (2017b). The impact of the neutral air turbulence

wavelength spectrum in the presence of charged dust particles is studied and the extension of the smaller wavelength electron





density fluctuation diffusion timescales in smaller wavelength is considered. While the coupling of neutral air turbulence
with mesospheric dusty plasma was investigated, the steady state situation and the balance of charging and diffusion time
scales on the plasma density fluctuation amplitude was overlooked. Therefore, the present study provides the first multi-
frequency observations of PMSE from 8 MHz up to 930 MHz (corresponding to electron density fluctuation wavelength $\sim$
m to 16 cm, respectively) as well as time evolution of plasma density fluctuations in numerical modeling of dusty plasma
irregularities within associated radar wavelengths. The diffusion and charging processes, electron and ion density variation,
and electron density fluctuation amplitude corresponding to the radar echoes are studied for a variety of background dusty
plasma parameters. The agreement between the numerical simulations and four radar frequency observations of the PMSE
source region simultaneously are presented and discussed.

## 2   Experimental observations

The data collected during 2012 and 2013 research campaigns at EISCAT HF Facility at Ramfjordmoen, Norway (69.6°N,
19.2°E), are presented in this paper. HF radar at $\sim$ 8 MHz (7.9 MHz) was built using the divided the HF heater array as
the HF radar transmitter. While 10 transmitters on Array 1 where used for heating the ionosphere by radiating vertically in
O-mode polarization at 6.77 MHz (an effective radiated power (ERP) of about 600 MW), simultaneously, the Facility operated
in radar mode radiating vertically using two transmitters at 7.953 MHz on Array 1. This transmission was cycled 48 s on
and 168 s off starting at 09:00 UT for the case of 10 June 2013 and 26 July 2013. The HF heater alternated between O-
and X-mode polarization. The HF radar transmission used a pair of 20 baud complementary codes (10 $\mu$s bauds and 1.5 km
range resolution). The receiving antenna O-mode gain was 25.2 dBi. The O-mode-transmitted pulses received using O-mode
polarization separation are used in this study. A complete measurement cycle took 80 ms. It should be noted that the HF radar
data presented in this paper are not calibrated for the D-region absorption. The HF radar RCS (radar cross-section) is used
in this paper to denote that the effect of ionospheric absorption is not taken into account. Echoes were received digitally and
combined in software to produce O or X polarization using Array 3 from two orthogonal linear polarizations. The Morro (the
MObile Rocket and Radar Observatory) radar at 56 MHz was used to measure the PMSE on the border line of HF and VHF
bands (Havnes et al., 2015). The Morro radar was established at EISCAT in 2008 and decommissioned in 2016 (Næsheim et
al., 2008). The EISCAT 224 MHz and 930 MHz radars monitored the mesospheric echoes in the same direction. The VHF
radars (56 MHz and 224 MHz) are not significantly affected by absorption.

Figure 1 shows the simultaneous PMSE observations with 4 radar frequencies. The UHF (930 MHz) radar observation is
shown on top panel and was stopped due to technical issue around 11:30 UT. The background electron density of the order
of $10^{11}$ m$^{-3}$ extends to 80 km altitude, where the PMSE associated dust/ice particles form, is observed. The VHF radar (224
MHz), the second panel from top, also represents the background electron density with a similar magnitude as those observed
with UHF radar. The VHF radar also shows simultaneous coherent PMSE that corresponds to electron density fluctuation
wavelength of $\sim$ 67 cm. The third panel shows the radar backscatter from the polar mesospheric clouds at 56 MHz. The
lower panel shows the HF PMSE at 7.9 MHz which is obtained using the Software Defined Radio Receiver (SDR) radar in



conjunction with the EISCAT HF facility. As observed in Figure 1, no PMSE is depicted at UHF (corresponding electron density fluctuation wavelength of 16.5 cm) during the time of operation. The VHF echoes appear between ∼ 80-87 km starting

around 10:30 UT. The VHF PMSE extends to altitudes lower than 85 km and becomes much stronger after 12:00 UT. While the overall shape of the PMSE layer looks very similar at 56 and 224 MHz, a clear extension and presence of the PMSE layer even before 10:30 UT can be seen in the Morro radar observations. The HF radar has received backscattered echoes from the beginning of the observations at 8:00 UT. The HF echoes show a much stronger pattern and extended over a wider altitude range in comparison with the 56 and 224 MHz echoes. Therefore the observations show consistency with the well

developed theory of neutral air turbulence which predicts an extension of turbulence to higher wavenumbers ($k$) (smaller irregularity wavelength $\lambda$) by including the reduced diffusion of fluctuations in the presence of heavy dust particles. This is mainly considered as the source of dusty plasma fluctuations and associated radar echoes (PMSE). Therefore, this will be subsequently investigated quantitavely using numerical simulations. The HF echoes appear at a higher altitude range between 85 km to 90 km (8:00 UT-11:00 UT). A slow descent in altitude to the ranges around 85 km is seen for times after 11:30 UT

which is in agreement with the observed 56 and 224 MHz echoes. As discussed in the introduction, the theoretical models are still incapable of characterizing the observed radar echoes in terms of strength and duration with realistic dusty plasma parameters corresponding to the PMSE source region.

Figure 2 shows a similar experimental set up to Figure 1 excluding the 930 MHz radar observations. A stable PMSE layer can be seen from the beginning of the observations at around 09:00 UT on 19 June 2013 at 56 MHz and 224 MHz. The overall

shape of the layers including the layering phenomenon between 9:50 UT and 11:20 UT as well as the curving structure around 12:30 UT can be clearly seen at both 56 and 224 MHz. The comparison between the two frequencies illustrates that the VHF echoes reach a very large amplitude at some times during the experiment and over a narrow altitude range (10:00 UT-11:00 UT). The heating modulation effects including the suppression of radar echoes after the HF heater turn-on and slow recovery to the background unperturbed echoes after a turn-off overshoot effect can be seen in 56 and 224 MHz echoes (Havnes et

al., 2015). Although the signature of HF pump modulation can be seen in the data, the overall intensity of the echoes at both frequencies remains intact. La Hoz et al. (2006) have also shown that a large increase in the electron temperature does not affect mesospheric neutral turbulent state as an external driver for PMSE. This will make it possible to use the data presented here as a benchmark for comparison with the numerical simulations presented in the following section in order to investigate neutral air turbulence effects on the formation of plasma irregularities. The overall range of presence in altitude and intensity

of 56 MHz echoes is higher in comparison with the 224 MHz echoes. The HF echoes are much stronger in comparison with 56 and 224 MHz, and cover a much wider altitude range. As can be seen in Figure 2, the 8 MHz echoes start around 84 km and extend to ∼ 91 km. Towards the end of experiment the lower bound of HF echoes reaches ∼ 80 km (after 11:00 UT). No layering structure is seen in 8 MHz echoes. It's noteworthy that the HF data are not calibrated for the lower ionospheric absorption which is appeared to be of the order of a few dBs (Senior et al., 2014).

Figure 3 includes the PMSE at all 4 radar frequencies available at EISCAT on 26 July 2013 between 9:00 UT to 13:00 UT. The UHF (930 MHz) echoes imply the background mesospheric conditions were quiet during this experiment. The UHF data shows a weak PMSE layer around 82 km starting at 11:40 UT to 12:00 UT. All other three radars used in the experiment show





the existence of PMSE at various strength. The 224 MHz radar show a discontinuous echoes center around 84 km. While the
lower VHF PMSE layers continue to 11:30 UT, a much stronger and double 224 MHz echo layer appear between 85 to 87 km.

Similar behavior of layering structure to Figure 2 is observed at 56 and 224 MHz, although the 8 MHz radar shows a continuous
layer of echoes with altitude. Overall intensity of HF echoes is much higher than 56 and 224 MHz. Such a difference can be seen
for 224 MHz in comparison with 930 MHz. This effect will be investigated throughout this paper using numerical simulations
of the perturbed mesospheric dusty plasma with different background parameters. This study aims at elucidating on the theory
of neutral air turbulence as the source of mesospheric plasma density fluctuations responsible for coherent radar echoes and in

the proximity of dust particles. The size and density of dust particles along with the initial amplitude of irregularities within
dust density as a result of coupling with neutral turbulence will be investigated.

## 3   Numerical simulations

One way of understanding the response of the electron density fluctuation amplitude to the background dust plasma parameters
is to use the analytical expression for the timescale of physical processes that actually affect the density fluctuations. In general

two processes of charging (electron/ion attachment to the dust particles) and plasma density diffusion determine the steady
state amplitude of fluctuations in the plasma density. The diffusion process tends to smooth out irregularities and can be
approximated for the natural PMSE layer ($T_e/T_i = 1$) by (Chen and Scales, 2005; Mahmoudian et al., 2011):

$$\tau_{diff} \approx \nu_{in} \left( \frac{\lambda_{irreg}}{2\pi v_{thi}} \right)^2 \frac{1}{\left( 1 + \left( 1 + \frac{z_{d0}n_{d0}}{n_{e0}} \right) \right)} \tag{1}$$

where $\nu_{in}$, $z_{d0}$, $n_{e,(d)}$ $\lambda_{irreg}$ and $v_{thi}$ are the ion-neutral collision frequency, charge density on the dust particles, electron (dust)

density, electron density irregularity wavelength and ion thermal velocity, respectively. According to the theoretical expression
of diffusion time scale this is mainly due to the dependency of diffusion time scales on $\lambda_{irreg}$.

The timescale for electron attachment onto the dust is approximated by

$$\tau_{chg} \approx \frac{1}{\sqrt{8\pi}r_d^2 v_{te0} e^{-4.1} n_{d0}} \tag{2}$$

While such simple theoretical expressions predict the dependency of radar echoes on the background dusty plasma parame-

ters, they are unable to predict the time evolution nor steady state amplitude of irregularities ($\delta n_e^2$) responsible for radar echoes.
Therefore, the computational model is incorporated in this paper to study the characteristics of radar echoes with respect to the
dusty plasma parameters. As can be seen in Eq. (1), the diffusion timescale depends on the $\lambda_{irreg}$. The numerical simulations
presented in this paper are associated with radar frequencies slightly different from the observations due to a limitation of the
model having a discrete spatial grid. The difference is of the order of a few centimeters in fluctuation wavelength, and are not

expected to change the physical processes and the results.



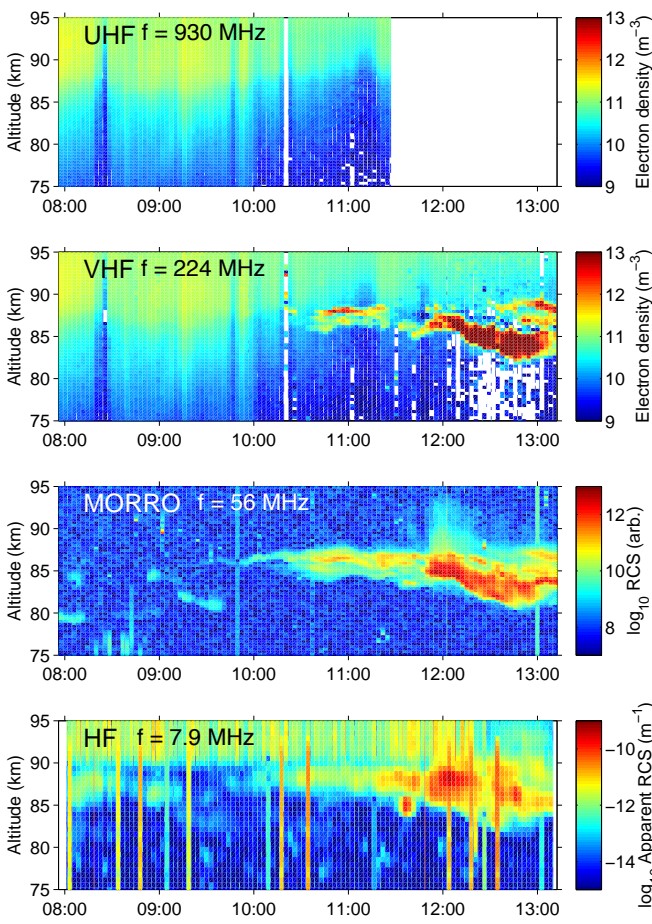

**Figure 1.** Simultaneous PMSE observations using 930 MHz, 224 MHz, 56 MHz, and 8 MHz radars, respectively from the top to the bottom, at EISCAT. The experiment was conducted on 12 July 2012 between 8:00 UT to 13:10 UT.

In order to investigate the time evolution of fluctuations in the plasma density due to coupling with neutral air turbulence, the dusty plasma model originally developed at Virginia Tech is used in this study (Scales, 2004; Mahmoudian et al., 2011). The model treats the plasma as a fluid including an arbitrary number of charged particles, neutral particle species and dust/aerosol particles which are modeled as particle in cell (PIC) (Birdsall and Langdon, 1991). Continuous charging model based on the Orbital-Motion-Limited (OML) approach has been used for the time varying charge on the dust particles. The summer mesopause temperature for both ions and electrons is taken to be $T_e = T_i = 150$K. The ion-neutral collision frequency is of order of $10^5$ s$^{-1}$. The electron density is assumed to be $2 \times 10^9$ m$^{-3}$. The numerical simulations presented in this paper includes



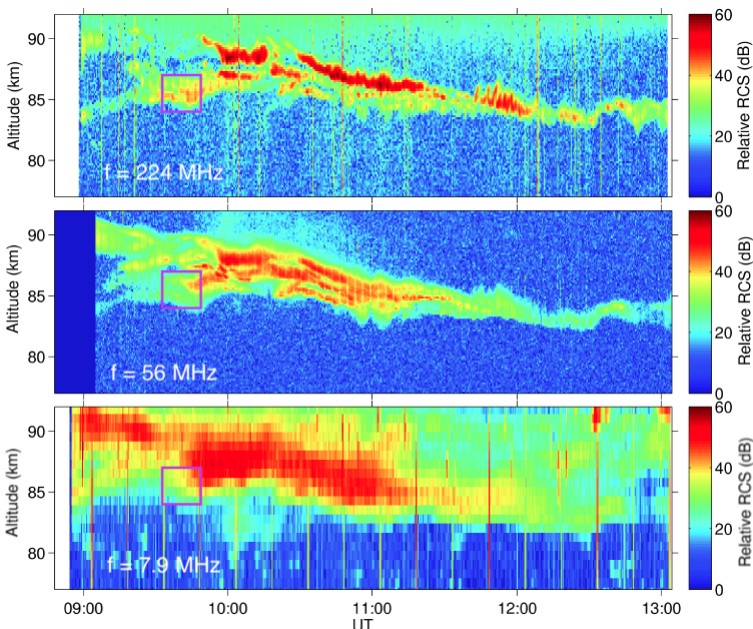

**Figure 2.** Simultaneous PMSE observations using 224 MHz (top), 56 MHz (middle), and 8 MHz (bottom) radars at EISCAT. The experiment was conducted on 19 June 2013 between 9:00 UT to 13:00 UT. The wavelength associated with each panel correspond to the radar frequency is shown. The electron density fluctuation wavelength are at the half of radar wavelength (Bragg scattering condition).

a wide range of dusty plasma parameters such as small and large dust particles as well as various dust densities. This is mainly to test the proposed theories such as fossil turbulence in order to explain the observed PMSE at a small wavelength where the

original neutral turbulence has disappeared due to kinematic viscosity and turbulence energy dissipation. A close comparison of computational results with the radar observations will be discussed in the following section to determine the most important dusty plasma parameters responsible for the long duration of small scale plasma density fluctuations in the PMSE region.

The time variation of 3 parameters in the simulations are are considered in this study. These parameters include electron density fluctuation amplitude squared ($\delta n_e^2$) equivalent to radar echoes, diffusion to charging time scales ($\tau_{diff}/\tau_{chg}$), and

time evolutions of electron and ion densities, which corresponds to the electron density depletion as a result of charging on to dust particles. The model ran to reach the steady state condition assuming the formation of dust/ice particle in the vicinity of mesospheric plasma. The $\tau_{diff}/\tau_{chg}$ value is a critical parameter that governs the time evolution and determines the steady state amplitude of electron density fluctuation amplitude (radar backscattered amplitude). The electron density is studied in order to investigate the electron depletion level as a parameter that also influences the backscattered radar signal as well as the

ion density to monitor the quasi-neutrality condition in the dusty plasma. The electron density fluctuation amplitude starts to increase initially due to dominant charging process. As the diffusion time scale becomes comparable to the charging timescale, it slows down the increase of electron irregularity amplitude until they reach steady state situation. Such a condition is achieved when the dust particles are saturated by the electron/ion charging currents. According to $\tau_{diff}/\tau_{chg}$, for small dust particles





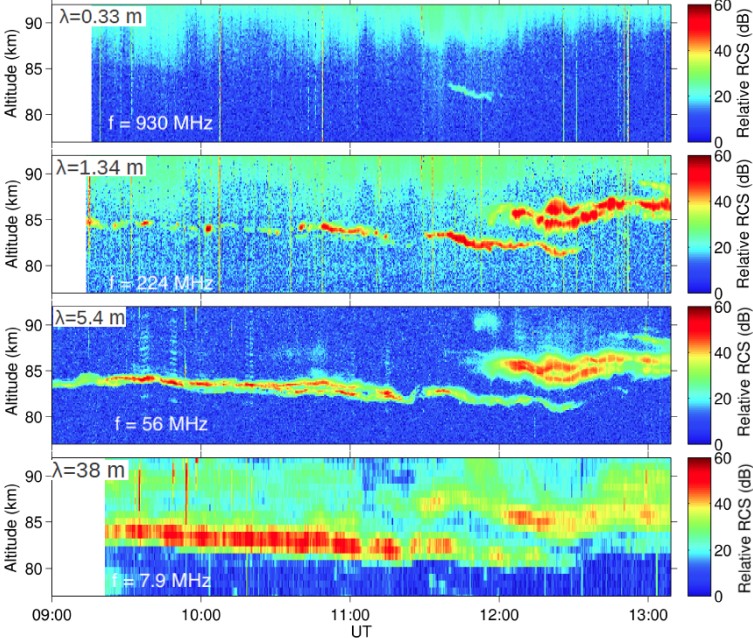

**Figure 3.** Simultaneous PMSE observations using 930 MHz, 224 MHz, 56 MHz, and 8 MHz radars, respectively from the top to the bottom, at EISCAT. The experiment was conducted on 26 July 2013 between 9:00 UT to 13:00 UT.

of 3 nm and 10 nm this condition is satisfied within 180 sec and $\sim$ 80 sec, respectively (Figures 4-7). The initial fluctuation
amplitude in the dust density ($\delta n_d/n_d$) which is caused by neutral air turbulence produces a footprint in the background density.
As will be discussed shortly, increase of $\delta n_d/n_d$ from 0.5 to 1 causes a substantial increase in $\delta n_e^2$. Therefore, this parameter
has a great impact on the corresponding radar echoes in different bands. The black, red, blue and green colors are used in
Figures 4-7 for the radar frequencies 7.3 MHz, 58 MHz, 234 MHz and 930 MHz, respectively.

As can be seen in Figures 4 and 5, for smaller dust radius of 3 nm the ion density reaches values higher than the equilibrium
condition due to low ion charging rate with respect to electron charging and in comparison with the production rate (photoion-
ization). For larger dust particles of 10 nm, the ion density reduces as a result of ion charging process on to the dust particles.
Such a reduction is of the order of 2 and 15% for dust densities of 10% and 70% ($n_d/n_{e0}$), respectively. One of the main
characteristics that can be seen in the electron density depletion at various wavelengths (radar frequencies) is the symmetri-
cal behavior. The average electron density (total depletion) is independent of radar frequency. The depletion level reaches $\sim$
20%, 35%, 12%, and 70%, in Figures 4-7, respectively. A very small discrepancy in the averaged electron density reduction
is seen for large dust particles of 10 nm and high dust density of 70% ($n_d/n_{e0}$) (Figure 7c). This is of the order of 2%, that is
negligible. This behavior excludes the idea that electron density depletion could contribute directly in the PMSE. Such effects
prove the coherent nature of the PMSE. Figures 4a, 5a, 6a, and 7a show the time evolution of $\delta n_e^2$, which is proxy for the radar


echoes. The right panel shows the corresponding radar echoes (amplitude $log_{10}N_e$ m$^{-3}$). The values are calculated assuming

$\delta n_d/n_d = 100\%$. For the solid curves associated with $\delta n_d/n_d = 50\%$, the values should be subtracted by $\sim$ -0.7 dB.

Another unique feature observed in the numerical simulations is the temporal evolution of $\delta n_e^2$. While for low densities (independent of dust radius) a slow increase in $\delta n_e^2$ amplitude to steady state value is observed (Figures 4a and 6a), for higher dust densities an overshoot effect and then a slow decrease to the steady state value is seen (Figures 5a and 7a). Such behavior is mainly due to high initial charging process associated with more dust particles present in the plasma. This phenomena can

be clearly seen in the $\tau_{diff}/\tau_{chg}$ plot for $r_d = 10$ nm, $n_d/n_{e0} = 70\%$ (Figure 7b). According to this figure, a sharp decrease in $\tau_{diff}/\tau_{chg}$ for values below $10^{-6}$ around 40 sec shows the dominant diffusion process that results in a slow decrease in $\delta n_e^2$ amplitude.

The temporal evolution of electron density fluctuation amplitude associated with four parameter regimes are examined in this paper. The four parameter set-ups are chosen based on the major parameters that could affect the radar echoes. According to the

charging and diffusion timescales that determine the steady sate value of radar echoes (electron density fluctuation amplitude at different wavelengths), dust radius $r_d$, dust density $n_d$, and dust density fluctuation amplitude $\delta n_d$ are considered to vary in the simulations. We assumed the heating modulation by the EISCAT HF transmitter has a minimal effect on the time evolution of radar echoes over 4 hours of observations ($T_e/T_i = 1$). The $\delta n_e^2$ and the corresponding radar echoes increases drastically with increasing $n_d$. A comparison of the results associated with the same dust radius (for example 3 nm) shows an increase

of $\delta n_e^2$ by a factor of 2 for all radar frequencies (irregularity wavelengths) as the $n_d$ increases from 90% to150%. A general empirical relationship for increase of $\delta n_e^2$ by a factor of $\sim (n_{d2}/n_{d1})^{(6/5)}$ is obtained based on the computational results for the same dust radius.

One of the main features that has been observed in the numerical results is that the $\tau_{diff}/\tau_{chg}$ for the radar frequencies 234 MHz and 936 MHz is on the same order. As can be seen in Figures 4-7, changing the background dusty plasma parameters

may vary the $\delta n_e^2$ which corresponds to the radar echoes, but the diffusion and charging time scales stay about the same. A small difference in the estimated amplitude of $\delta n_e^2$ for similar dust parameters associated with 224 MHz and 930 MHz radars contradicts the observational data. Considering that the observed UHF echoes is expected to be in a similar altitude range as other radar frequencies, the background dust parameters such as $r_d$ and $n_d$ should be the same in the simulations. Therefore, a close comparison between the $\delta n_e^2$ for 234 MHz and 936 MHz from the simulations with the same $r_d$ and $n_d$, reveals a

very small difference in the amplitude. Such numerical prediction requires the observation of PMSE at 930 MHz. This is not consistent with the observations. The numerical estimation of $\delta n_e^2$ for reduced $\delta n_d/n_d$ values from 1 to 0.5 shows that the expected radar echoes are estimated to decrease at least 1 dB. This could explain the non observation of UHF PMSE due to dissipation of fluctuations at such a short wavelength.

## 4   Summary and conclusion

The previous studies have shown that the small-scale electron density fluctuations produced through coupling of neutral air turbulence and charging process of electrons onto irregular dust density may be extended to higher $k$ values (corresponding



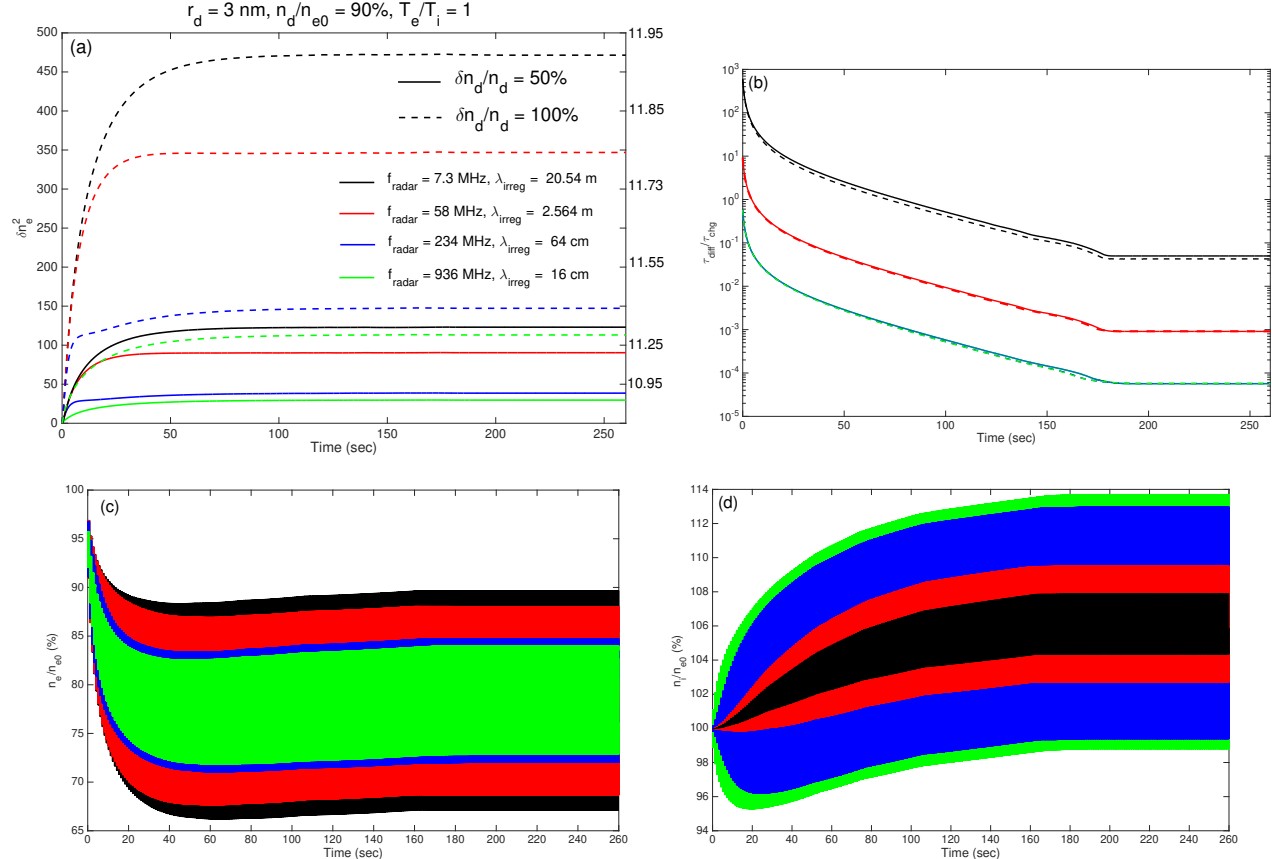

**Figure 4.** Numerical results associated with radar frequencies of 7.3 MHz ($\lambda_{irreg} = 40.96m$), 58 MHz ($\lambda_{irreg} = 5.12m$), 234 MHz ($\lambda_{irreg} = 1.28m$), and 936 MHz ($\lambda_{irreg} = 32cm$). the mesospheric plasma parameters are $r_d = 3$ nm, $n_d/n_{e0} = 90\%$, $T_e/T_i = 1$. The dashed lines correspond to the the initial dust fluctuation amplitude with respect to the background dust density $\delta n_{d0}/\delta n_d = 100\%$, and solid lines denote $\delta n_{d0}/\delta n_d = 50\%$. (a) shows the time evolution of electron density fluctuation amplitude squared ($\delta n_e^2$), which is a proxy for the radar echoes. (b) represents the diffusion to charging time scales associated with each radar echoes. (c) the normalized electron density variation due to charging process on to the dust particles. (d) time evolution of the normalized ion density variation to the background electron density.

to smaller wavelength). Such effect could explain the coherent PMSE echoes observed at higher radar frequencies. While the increase of the Schmidt number to values much greater than unity (in the absence of dust particles) and extension of the viscous cut-off have been shown to some extent, no clear observational and computational modeling of PMSE at various frequency
bands has been introduced.

This paper provides the first simultaneous observations of the PMSE source region with 4 ground-based radars. The radar frequencies 8 MHz, 56 MHz, 224 MHz, and 930 MHz corresponding to Bragg scatter (coherent) electron density fluctuation wavelengths of $\lambda_{irreg} = 40.96$ m, 5.12 m, 1.28 m, and 32 cm, respectively, were employed. Such a wide range of radar



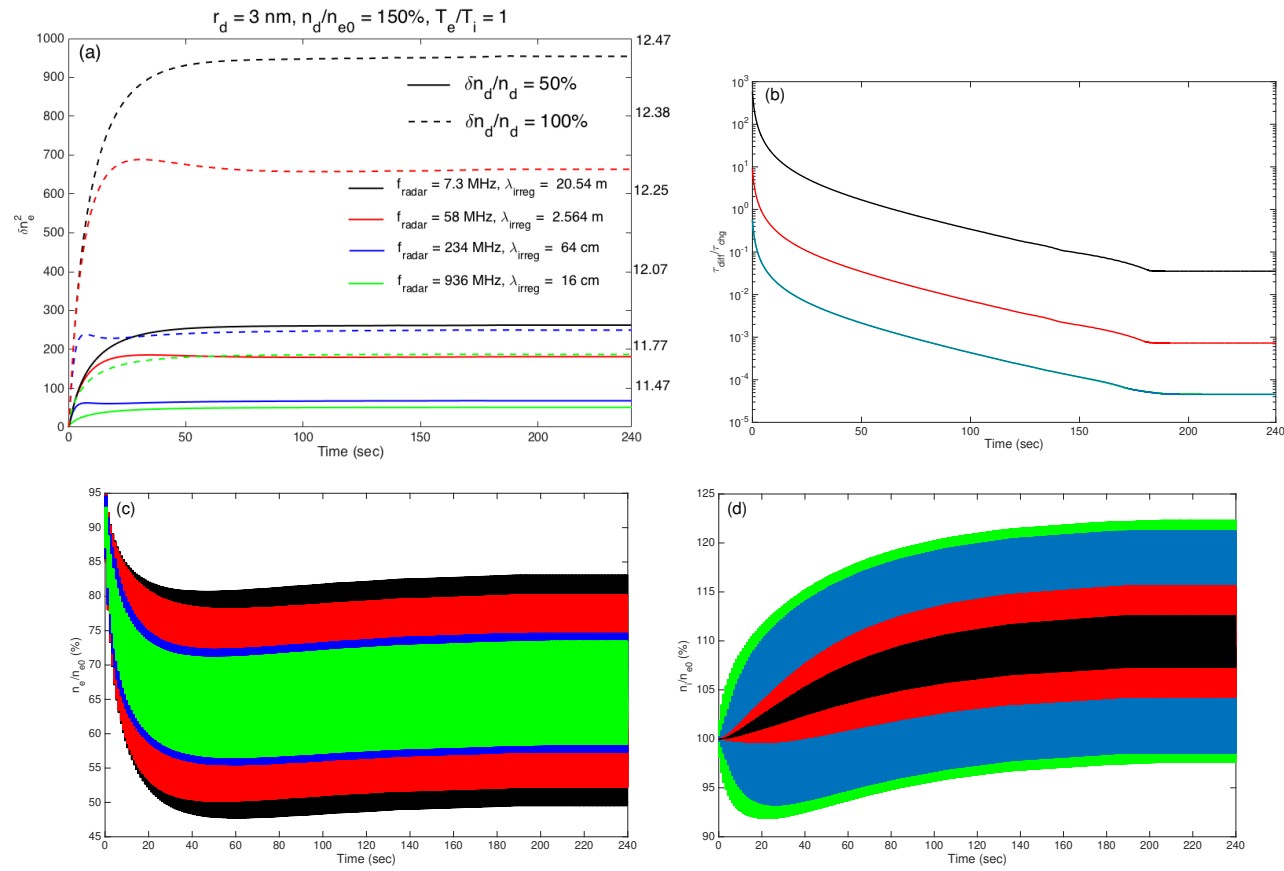

**Figure 5.** Similar to Figure 4 for $r_d = 3$ nm, $n_d/n_{e0} = 150\%$, $T_e/T_i = 1$.

frequencies cover the irregularity wavelength regimes associated with the neutral air turbulence theory. This paper investigates

the time evolution and steady state amplitude of the electron density fluctuation ($\delta n_e^2$) in the presence of naturally perturbed mesospheric dust layers. The numerical calculation of $\delta n_e^2$ could be compared quantitively with the experimental observations of coherent radar echoes at various frequency band. Since the initial modulation of the dust density profile by the background neutral density profile has a great impact on the corresponding electron density fluctuation amplitude and radar echoes, two values of 0.5 and 1 are assumed for $\delta n_d/n_d$ parameter. Then plasma processes including electron/ion attachments to the

background dust particles as well as electron density diffusion are allowed to develop and reach a steady state condition. Several background dusty plasma parameters including dust density and dust radius are allowed to vary in order to determine the corresponding amplitude of electron density fluctuation and the associated radar echoes at different wavelengths. Another important parameter that was investigated is the initial amplitude of dust density irregularities produced through neutral air turbulence ($\delta n_d/n_d$). In the case of larger dust particles of the size of 10 nm, the steady state condition is satisfied at much

shorter time in comparison with 3 nm dust particles. In this case the stable fluctuation amplitude is achieved within 80 sec from

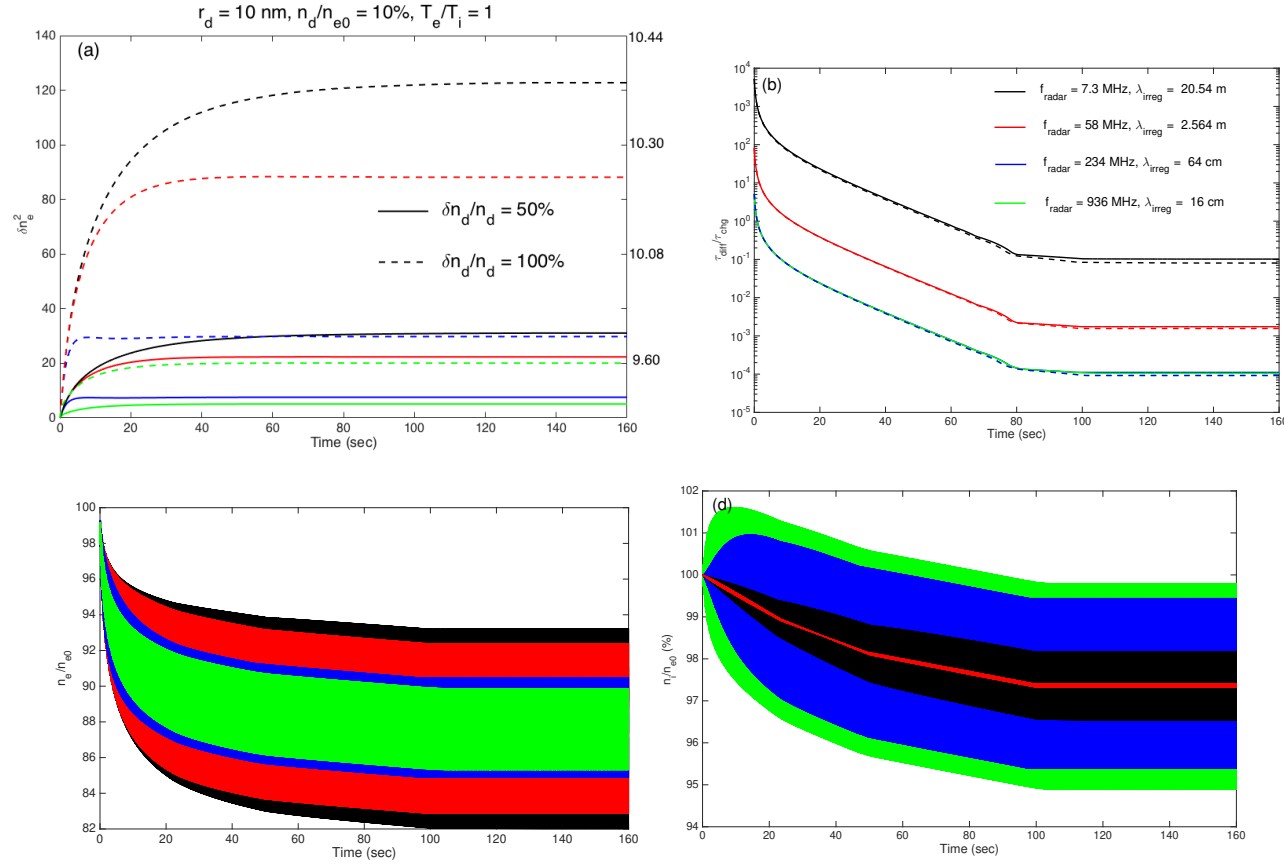

**Figure 6.** Similar to Figure 4 for $r_d = 10$ nm, $n_d/n_{e0} = 10\%$, $T_e/T_i = 1$.

the equilibrium condition (quasi-neutral plasma in the absent of dust particles). This parameter manifests itself in the steady state amplitude of electron density fluctuations. An empirical relationship for the $\delta n_e^2$ with dust density with the same dust radius has been obtained using the numerical results. It has been shown that $\delta n_e^2$ increases by a factor of $(n_{d2}/n_{d1})^{(6/5)}$.

One of the main features observed in the numerical simulations is that the electron density depletion is independent of 245 the fluctuation wavelength and only varies with the background dust parameters. This effect validates the coherent scattering mechanism considered as the major source for plasma density irregularity generation. Therefore, it has been shown in this paper that the direct role of electron density depletion on corresponding radar echoes at various frequency bands is not possible, unlike the previous conclusion by Varney et al. (2011). In fact, the electron density fluctuation amplitude plays the major role in the coherent scattering process. The numerical results associated with various background dusty plasma is considered in order to 250 investigate the 8, 56, 224, and 930 MHz PMSEs.

In-situ rocket and radar observations have proved that neutral air turbulence by itself has a minor effect on the creation of small scale plasma irregularities in the PMSE region. The rocket measurements have shown a sharp cut off of fluctuations



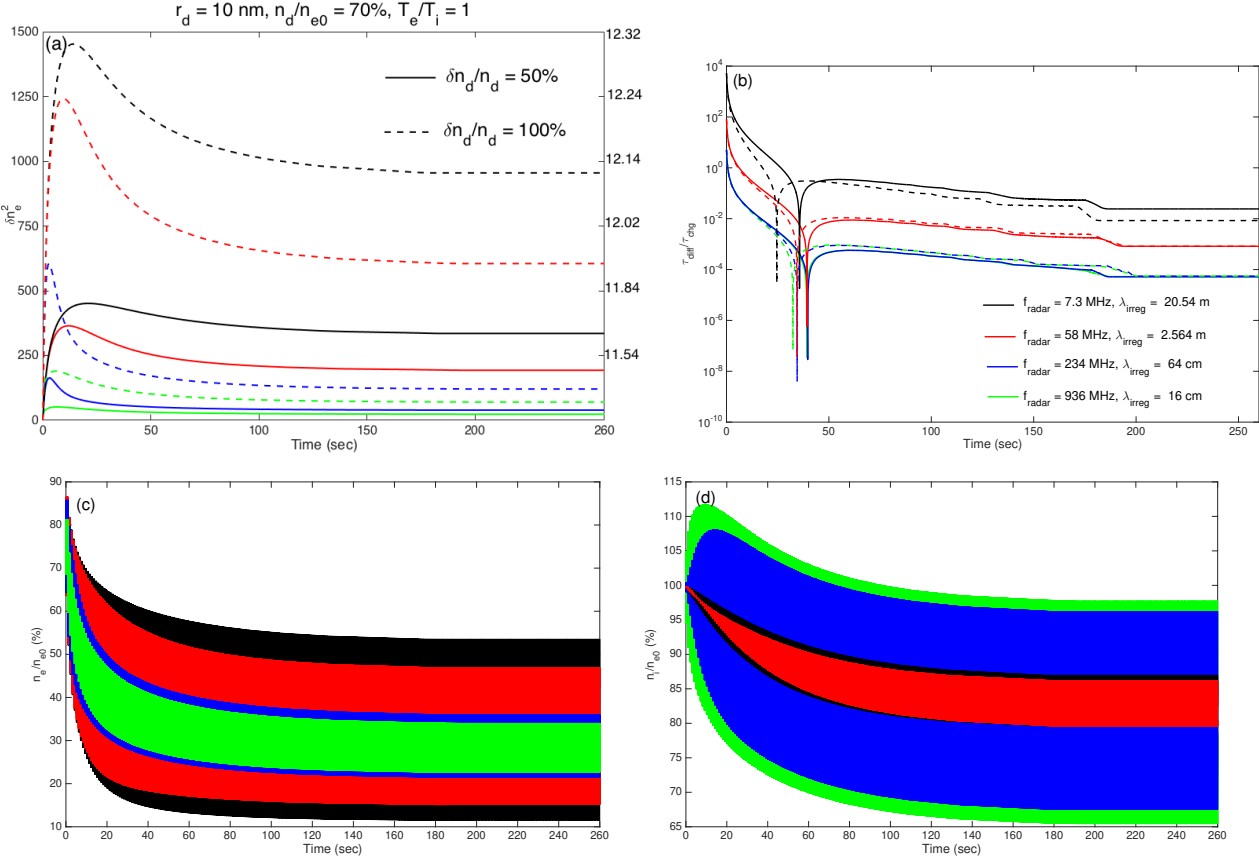

**Figure 7.** Similar to Figure 4 for $r_d = 10$ nm, $n_d/n_{e0} = 70\%$, $T_e/T_i = 1$.

of the order of a few tens of meters ($\sim 24$ m) (Rapp and Lubken, 2003). The long lifetime of PMSE below freezing altitude and even in the absence of neutral air turbulence have been attributed to frozen structure produced initially through turbulence

advection and due to reduced diffusivity. The typical power spectrum of turbulence motion has two distinct parts. The first part represents a tracer with a wavenumber dependence of $k^{-5/3}$, which is known as the inertial subrange. The power spectrum continues with a second part where power spectrum is proportional to $k^{-1}$. Such spectral variation predicts a near zero power amplitude for Bragg scale of $\sim 5$ m associated with $f_r \sim 56$ MHz due to dominant molecular diffusion. In general kinematic viscosity (diffusion of momentum) and turbulent energy dissipation rate (dissipation of turbulence energy to heat) are the two

main parameters that affect the minimum scale of irregularities in the inertial subrange. The previous works on fossil turbulence have shown the importance of dust particle size as the main parameter that controls the viscosity and minimum steady scale of irregularities. Particles as large as 10 nm-15 nm around 85 km were proposed to justify the extended range of PMSE observations (Rapp and Lubken, 2003; La Hoz et al., 2006). The numerical simulations presented in this paper demonstrate the





major role of dust density along with dust radius as the two main parameters that determine the general behavior of turbulence
power spectrum.

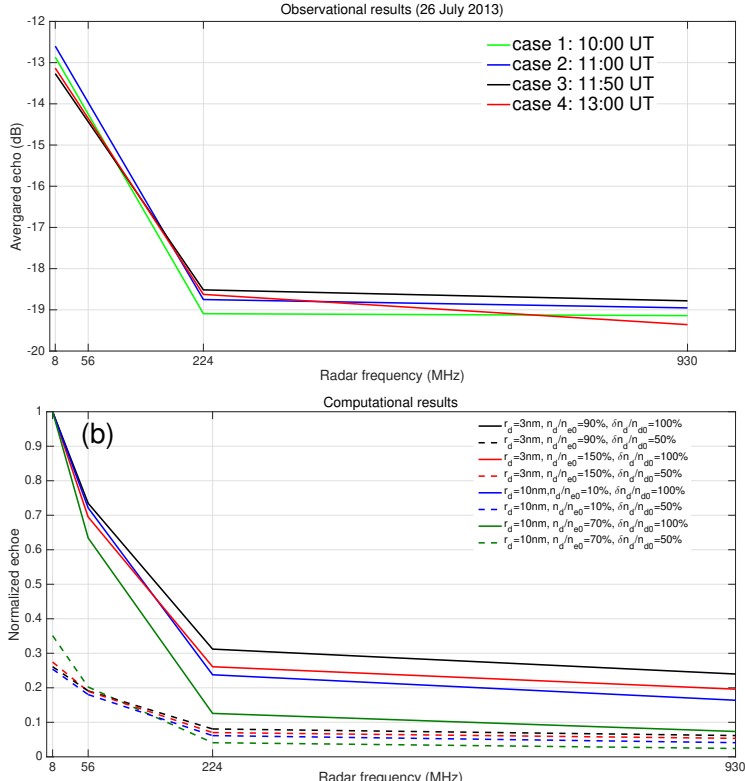

**Figure 8.** a) Averaged power echoes over the HF PMSE altitude range (in arbitrary unit) and associated with 26 July 2013, in 4 cases at 10:00 UT, 11:00 UT, 11:50 UT, 13:00 UT. b) Numerical results associated with Figures 4-7 and corresponding to the variation of the radar echo strength with dust parameters.

The general correlation of the shape of the PMSE at different frequencies obtained through the observations reveal the existence of the plasma density fluctuations and associated structure over the same altitude range. The high resolution of observations of the turbulence region probed with with 8 MHz, 56 MHz, 224 MHz, and 930 MHz radars show for the first time that fossil turbulence theory can be applied to the formation of PMSE. As discussed, while theoretical models fall short in
predicting the dusty plasma parameters associated with the observed PMSEs, a computation model is used in this paper in order to determine the background parameters as well as provide an explanation for the non-existence (strong weakening) of PMSE between 224 MHz and 930 MHz. According to the numerical results the dust radius 3 nm with density of 150 percent with respect to the background electron density ($n_e = 2 \times 10^9$ m$^{-3}$) can produce radar echoes of the order of 12.5 dB ($log10N_e$) for $f_r = 7.3$ MHz. This shows a close agreement with the experimental observations presented in Figure 1. Figure 8a presents
the actual amplitude of radar echoes over the HF PMSE altitude range (80.91 km to 91.4 km) and associated with 26 July



2013, in 4 cases at 10:00 UT, 11:00 UT, 11:50 UT, 13:00 UT. Figure 8b also provides a summary of numerical results of radar echo associated with Figures 4-7 and corresponding to the variation of the radar echo strength with dust parameters. A close comparison between the two figures show a good correlation in regards to strength consistency as the radar frequency increases from 8 MHz to 224 MHz and 930 MHz. Such averaging is essential to determine the effectiveness as well as durability of the

original turbulence in the presence of dusty plasma to generate coherent echoes from 8 MHz to 930 MHz (although it is very rare to observe PMSE at 930 MHz). Larger dust particles of 10 nm with smaller densities are also taken into account to examine the hypothesis of the role of larger dust particles in the PMSE formation. The numerical results has shown that for the similar dust parameters ($r_d$ and $n_d$), the expected radar echoes ($\delta n_e^2$) is very close for 224 MHz and 930 MHz. It has been illustrated that the reduction of dust density fluctuation amplitude ($\delta n_d/n_d$) by a factor of 2 would reduce the expected radar echoes by 1

dB. The overall drop in the PMSE amplitude shows a similar drop as the $\delta n_d/n_d$ reduces from 1 to 0.5 for all dust parameters presented in Figures 4-7. Therefore, although according to the numerical results the estimated PMSE amplitude is similar for 224 MHz and 930 MHz, the $\delta n_d/n_d$ plays a critical role in the steady state amplitude of irregularities and associated PMSE. This validates the non observation of UHF PMSE and can also be implemented to guide the previous theories used to predict the neutral air turbulence spectrum in the presence of dust particles.

*Acknowledgements.* EISCAT is an international association supported by research organizations in China (CRIRP), Finland (SA), Japan (NIPR and STEL), Norway (NFR), Sweden (VR), and the United Kingdom (NERC). The authors would like to thank Andrew Senior for his help in the data processing.

The data presented in this paper can be downloaded from the EISCAT online database at https://www.eiscat.se/scientist/data/



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
