# Peer review of "Neutral air turbulence in the mesosphere and associated polar mesospheric summer echoes (PMSEs)"

_Annales Geophysicae, 2020_

## Referee Comment (RC1) · Anonymous Referee #1 · 4 Jan 2021

This paper presents the simultaneous PMSE observations at four radar frequenciesand a comparison to a model of the PMSE formation. The presented observations are new and interesting and the manuscript may include new scientific results. A meaningful review of the manuscript is only possible after additional work and revision.

The authors present an interesting set of new observation, but unfortunately the organization and writing of the manuscript makes it hard for the reader to follow the presentation and to understand its conclusions: The paper includes a lengthy introduction, but it is not evident from the introduction what exactly the authors want to study. The numerical model that the authors apply is not sufficiently described. The paper

includes a comparison of normalized echo powers measured with different radars, but the applied normalization is not described. The observations are made during articficial heating. It is not sufficientlu described how the heating is included in the numerical model that the authors compare to the observations. The conclusion section includes mainly discussion and it is difficult for the reader to find a conclusion to the work. The language is not precise and the text is not well structured.

Minor comments The labels in figures 4 – 8 are too small. Check spelling of ÂńLue-bkenÂż.

---

## Referee Comment (RC2) · Anonymous Referee #2 · 18 Jan 2021

The paper is devoted to investigation of polar mesospheric summer echoes (PMSE) and aims at comparing simulations with radar observations. The title suggests that the main subject of this study will be the neutral air turbulence. The abstract further defines several key points which should be addressed in this work:

1. "four radar frequency observations of the PMSE region under varying neutral air turbulence conditions."

2. "effect of neutral air turbulence on the generation and strength of PMSE" as a function of "dust parameters"

[Figure]

3. "neutral air turbulence in presence of heavy dust particles, can largely explain the observed radar cross-section at four radar frequencies"

4. "effect of initial turbulence amplitude" as dependence on "of various dust parameters"

5. "Several key parameters in dusty plasma responsible for the PMSE observations are determined"

The paper is badly organized, difficult to read, and contains vast of misleading formulations (wrong definitions and descriptions). As a result, it is difficult to follow the authors ideas and interpretations.

I did not find satisfactory answers on most of the defined above problems and cannot suggest a simple way of improving this manuscript. Therefor I suggest rejection.

General structure of the manuscript must be improved by separating model description, simulation results, its comparison with measurements, and conclusions. Currently everything is mixed in two sections 3 and 4.

PMSE study at three radar frequencies can be found in work of *Rapp et al.* (2008), where it was quantitatively shown based on measurements, that turbulence in presence of heavy aerosols well explains PMSE strength and frequency (Bragg scale) dependence. Also, scattering of radio waves as dependence on turbulence parameters was addressed in detail by *Lübken* (2014). Therefore, authors must show what is new in their paper.

However, I may admit that if measurements shown here were made in a true common volume (which is not clearly said in the manuscript) they are quite interesting and I would encourage authors to submit a more consistent and detailed study based on these data.

In what follows I give some more specific comments.

- To point 1 above: I did not understand from the article how the "turbulence conditions" were "varying" during measurements and how it was inferred from those measurements. I've got an impression, that turbulence was not estimated from the measurements, however it might be possible, e.g., from spectral width of the radar measurements.

- The same is also valid for the second goal claimed in the abstract (point 2 above). Also, as I understand, the authors use "dust density fluctuation amplitude" as a measure of turbulence intensity. However, this relation is not justified or explained in the paper (e.g., at least by assessment of time constant). Also, I did not find, "effect on PMSE generation", e.g., range of "dust parameters" and turbulence intensity when PMSE is not generated, or threshold values, whatsoever.

- In abstract stated: "neutral air turbulence in presence of heavy dust particles, so-called fossil turbulence" (point 3 above). It is wrong formulation. You can name it dusty turbulence, but not fossil. Fossil turbulence can be characterized by presence of turbulent-like structures in scalar fields whereas the velocity field became laminar. Also, as mentioned above, this point was much better addressed by *Rapp et al.* (2008).

- To point 4 above: The term "various dust parameters" must be explained better, e.g., by a table of these parameters, or by sub-sectioning the paper explaining these dependencies subsequently. Also practical meaning of the "initial turbulence amplitude" in this study is unclear. Is it a pure feature of simulations or it can be used for interpretation of measurements?

- The last point in the list of goals mentioned above is not answered in this paper as well. However, this is already answered in numerous works published so far.

**References**

Lübken, F.-J., Turbulent scattering for radars: A summary, *Journal of Atmospheric and Solar-Terrestrial Physics*, *107*, 1 – 7, doi:https://doi.org/10.1016/j.jastp.2013.10.015, 2014.

Rapp, M., I. Strelnikova, R. Latteck, P. Hoffmann, U.-P. Hoppe, I. Häggström, and M. Rietveld, Polar Mesosphere Summer Echoes (PMSE) studied at Bragg wavelengths of 2.8 m, 67 cm, and 16 cm, *J. Atmos. Solar-Terr. Phys.*, *70*, 947–961, doi:10.1016/j.jastp.2007.11.005, 2008.

---

## Author Comment (AC1) · 8 Feb 2021

This paper presents the simultaneous PMSE observations at four radar frequenciesand a comparison to a model of the PMSE formation. The presented observations are new and interesting and the manuscript may include new scientific results. A meaningful review of the manuscript is only possible after additional work and revision. The authors present an interesting set of new observation, but unfortunately the organization and writing of the manuscript makes it hard for the reader to follow the presentation and to understand its conclusions: The paper includes a lengthy introduction, but it is not evident from the introduction what exactly the authors want to study. The numerical model that the authors apply is not sufficiently described. Discussion paper includes a comparison of normalized echo powers measured with different radars, but the applied normalization is not described. The observations are made during articficial heating. It is not sufficientlu described how the heating is included in the numerical model that the authors compare to the observations. The conclusion section includes mainly discussion and it is difficult for the reader to find a conclusion to the work. The language is not precise and the text is not well structured.

The authors agree with the referee that the paper could benefit from reorganization. Therefore, author take this comment into full consideration by rewriting the vague parts of the paper as well as reorganizing the paper through separating the model description, comparison between observational and numerical results, clearly explain the difference with the previous work and novel results provided by this paper. In short summary, the present paper provided the first common volume observations of PMSE source region with 4 radars including the 7.9 MHz radar for the first time in such study. Having a low frequency radar corresponding to high wavelength in the fluctuations is critical to make a solid conclusion to explain the source of irregularities as well as provide an exact estimation of background dusty plasma parameter (including dust radius and density) to achieve the radar echoes at level observed at 4 frequencies. Unlike the similar past works that had limitation in the experiments including uncorrelation in the probed region by different radars, no observations in the HF band, as well as using simple theories of neutral turbulence in the presence of dust particles that required high Schmidt number associated with large dust particles, the present work uses the full computational model to study microphysics of this region and evolution of radar echoes in response to the background dust parameters. Therefore, the finding of the paper has shown for the first time that smaller dust particles could also explain the radar echoes. Moreover, estimation of background dust parameters along with neutral turbulence is another advantage of the present work.

 Minor comments
The labels in figures 4 – 8 are too small.
Check spelling of ÂnLue- ´ bkenÂz.
Authors have included the minor comments into the paper.

---

## Author Comment (AC2) · 8 Feb 2021

1. "four radar frequency observations of the PMSE region under varying neutral air turbulence conditions."
2. "effect of neutral air turbulence on the generation and strength of PMSE" as a function of "dust parameters"
3. "neutral air turbulence in presence of heavy dust particles, can largely explain the observed radar cross-section at four radar frequencies"
4. "effect of initial turbulence amplitude" as dependence on "of various dust parameters"
5. "Several key parameters in dusty plasma responsible for the PMSE observations are determined"

The paper is badly organized, difficult to read, and contains vast of misleading formulations (wrong definitions and descriptions). As a result, it is difficult to follow the authors ideas and interpretations. I did not find satisfactory answers on most of the defined above problems and cannot suggest a simple way of improving this manuscript. Therefor I suggest rejection. General structure of the manuscript must be improved by separating model description, simulation results, its comparison with measurements, and conclusions. Currently everything is mixed in two sections 3 and 4. PMSE study at three radar frequencies can be found in work of Rapp et al. (2008), where it was quantitatively shown based on measurements, that turbulence in presence of heavy aerosols well explains PMSE strength and frequency (Bragg scale) dependence. Also, scattering of radio waves as dependence on turbulence parameters was addressed in detail by Lübken (2014). Therefore, authors must show what is new in their paper. However, I may admit that if measurements shown here were made in a true common volume (which is not clearly said in the manuscript) they are quite interesting and I would encourage authors to submit a more consistent and detailed study based on these data. In what follows I give some more specific comments.

As the authors mentioned in his/her comments, our work (the present paper) is the first true common volume observations of PMSE source region with 4 radars. This is one of the main advantages of the present data in comparison with the previous works cited by the referee (such as Rapp et al., 2007) that probed different region of PMSE (130 km difference in PMSE location). Another main advantage of the present work is including the HF PMSE observations at 7.9 MHz (corresponding to 20 m wavelength). This is critical to make a correct judgment on the applied theories such as neutral turbulence with high Schmidt number.
There are numerous in-situ rocket observations in recent years that have probed PMSE source region in different months of the year, they unanimously have shown that the particles in the region are much smaller than what claimed in the conclusion of the paper cited by the referee. The referee argued in the referenced paper (Rapp et al., 2007) that dust particles of the size of 20 nm are required to get the best agreement with the observations and theoretical model.

Almost all recent in-situ rocket observations of PMSE source region in different months have shown dust/ice particles at much smaller radius are responsible for the radar echoes. Moreover AIM satellite observations have shown the similar concept. The present paper uses the state-of-the-art numerical model capable of simulating the PMSE source region including all physic processes considered. The results presented in the paper have shown that even small dust particles are capable of producing the radar echoes at the same level as those observed in the experiment. Therefore, the proposed model and the novel observational results presented in this paper are advancement to the field and previous works.

The authors agree with the referee that the paper could benefit from reorganization. Therefore, author take this comment into full consideration by rewriting the vague parts of the paper as well as reorganizing the paper through separating the model description, comparison between observational and numerical results, clearly explain the difference with the previous work and novel results provided by this paper.

• To point 1 above: I did not understand from the article how the "turbulence conditions" were "varying" during measurements and how it was inferred from those measurements. I've got an impression, that turbulence was not estimated from the measurements, however it might be possible, e.g., from spectral width of the radar measurements.

The experiments conducted in different days and years that impose the variation of turbulence condition. Even the radar echoes presented in the paper clearly show the variation in the strength of radar echoes in associated frequency bands.

• The same is also valid for the second goal claimed in the abstract (point 2 above). Also, as I understand, the authors use "dust density fluctuation amplitude" as a measure of turbulence intensity. However, this relation is not justified or explained in the paper (e.g., at least by assessment of time constant). Also, I did not find, "effect on PMSE generation", e.g., range of "dust parameters" and turbulence intensity when PMSE is not generated, or threshold values, whatsoever.

The referee mentioned two different thing that are totally un correlated. The initial amplitude of dust density fluctuations in the dust density is considered to be 50 or 100 percent with respect to the background dust density. The values used in this paper are based on recent observations and available theories. The model then let the physical processes including the competing charging and diffusion process to advance. The charging on to the dust particles intends to increase the electron density fluctuation amplitude. The diffusion process tends to have an opposite impact. Therefore, the final amplitude of electron density fluctuation that corresponds to the radar echoes is calculated by letting the simulation to reach the steady state. In other words, "dust density fluctuation amplitude" is just an input parameter that could reflect the strength of the turbulence. But what the model calculates and matters the most is the electron density fluctuation amplitude in presence of various dust density and radius.

• In abstract stated: "neutral air turbulence in presence of heavy dust particles, socalled fossil turbulence" (point 3 above). It is wrong formulation. You can name it dusty turbulence, but not fossil. Fossil turbulence can be characterized by presence of turbulent-like structures in scalar fields whereas the velocity field became laminar. Also, as mentioned above, this point was much better addressed by Rapp et al. (2008).

We appreciate the referee's comment on using the term Fossil turbulence. We corrected this for clarification. We have explained the weaknesses of the reference proved by the referee. We have also mentioned a detailed comparison as well as the advantages of our model in comparison with the previous studies.

• To point 4 above: The term "various dust parameters" must be explained better, e.g., by a table of these parameters, or by sub-sectioning the paper explaining these dependencies subsequently. Also practical meaning of the "initial turbulence amplitude" in this study is unclear. Is it a pure feature of simulations or it can be used for interpretation of measurements?

A new table will be added to the paper to explain the dust parameters used in the simulations. The criteria for using such parameters have been provided in the text. The initial turbulence amplitude has been explained in the response to the previous comments. It can be definitely used with the common volume observations at 4 frequencies to interpret the measurements.

 • The last point in the list of goals mentioned above is not answered in this paper as well. However, this is already answered in numerous works published so far.

References Lübken,
F.-J., Turbulent scattering for radars: A summary, Journal of Atmospheric and SolarTerrestrial Physics, 107, 1 – 7, doi:https://doi.org/10.1016/j.jastp.2013.10.015, 2014.

Rapp, M., I. Strelnikova, R. Latteck, P. Hoffmann, U.-P. Hoppe, I. Häggström, and M. Rietveld, Polar Mesosphere Summer Echoes (PMSE) studied at Bragg wavelengths of 2.8 m, 67 cm, and 16 cm, J. Atmos. Solar-Terr. Phys., 70, 947–961, doi:10.1016/j.jastp.2007.11.005, 2008.

The weaknesses of the above-mentioned references have been pointed out in our response. Moreover, the above-mentioned references admit the limitation of their work. It has been clearly stated that ultimate proof of their concept (Schmidt number between 2500 to 5000 which requires as large as 20 nm) requires direct measurement of ice particle sizes in a PMSE environment. Our study provides the first common volume observations of PMSE source region with 4 radars including the 7.9 MHz radar for the first time in such study. Having a low frequency radar corresponding to high wavelength in the fluctuations is critical to make a solid conclusion to explain the source of irregularities as well as provide an exact estimation of background dusty plasma parameter (including dust radius and density) to achieve the radar echoes at level observed at 4 frequencies. Unlike the similar past works that had limitation in the experiments including uncorrelation in the probed region by different radars, no observations in the HF band, as well as using simple theories of neutral turbulence in the presence of dust particles that required high Schmidt number associated with large dust particles, the present work uses the full computational model to study microphysics of this region and evolution of radar echoes in response to the background dust parameters. Therefore, the finding of the paper has shown for the first time that smaller dust particles could also explain the radar echoes. Moreover, estimation of background dust parameters along with neutral turbulence is another advantage of the present work.

---

## Author Comment (AC3) · 11 Feb 2021

The comment was uploaded in the form of a supplement:
https://angeo.copernicus.org/preprints/angeo-2020-81/angeo-2020-81-AC3-supplement.pdf

---

## Referee Comment (RC3) · Anonymous Referee #3 · 31 May 2021

Review of the manuscript titled "Neutral air turbulence in the mesosphere and associated polar mesospheric summer echoes (PMSEs) " by Mahmoudian et al.,

General Remarks

The present study reports the multi-frequency radar (930, 224, 56 and 7.9 MHz) observations of Polar Summer Mesospheric Echoes (PMSE) using EISCAT observations. The authors by employing the numerical simulations attempt to explain the physical mechanism responsible for the observed coherent radar echoes. The numerical simulations include the time evolution of electron density perturbations, which are respon-
sible for observed radar echoes, in the presence of dust layers in the mesosphere. Various dust parameters such as size, density and initial turbulence amplitudes are varied to estimate the electron density fluctuations. The results show that neutral air turbulence modulated dust particles known as fossil turbulence is responsible for the PMSE observed at four radar frequencies. This is the first time that results from radars operating at four frequencies are simultaneously employed to study the PMSE along with numerical simulations. The results discussed in the manuscript are of interest to Annals of Geophysicae community and I therefore recommend it for the publication. However, the authors have to implement the following suggestions before the manuscript becomes acceptable for publication.

Specific Comments

1.The units of the radar intensity maps are different for figure 1 as compared to figure 2 and 3. Authors have to change the units such that all the figures are comparable.

2.How the neutral turbulence is related to the dust fluctuations is not clear. Authors have to discuss whether the spectrum of neutral turbulence and the dust particle fluctuations are same or not?

3.How the authors explain the absence of echoes at 930 MHz, based on fossil turbulence theory? The explanation given by the authors should be substantiated with further discussion.

4.The authors state that "Fluctuations in dusty plasma may also be generated by "fossil turbulence" when neutral air turbulence is absent". However they discuss the coupling of neutral turbulence and initial amplitude of irregularities within dust density. If fossil turbulence forms in the absence of neutral turbulence then how the coupling between them is justified.

5.The discussion of the results is not very coherent and there are repetitions. Authors have to carefully go through the manuscript and try to avoid repetitions and firm up the

discussion.

Minor Comments line 145: evolution nor steady state → evolution of steady state line 191:low density → low dust density There is a scope for improving the English Grammar in the manuscript.

---

## Author Comment (AC4) · 12 Jun 2021

The authors would like to thank the referee for his/her valuable comments on the paper. The detailed response to the reviewer comments is provided below. The revised version of the manuscript will be prepared once the final decision is made.

Review of the manuscript titled "Neutral air turbulence in the mesosphere and associated polar mesospheric summer echoes (PMSEs) " by Mahmoudian et al., General Remarks The present study reports the multi-frequency radar (930, 224, 56 and 7.9 MHz) observations of Polar Summer Mesospheric Echoes (PMSE) using EISCAT observations. The authors by employing the numerical simulations attempt to explain the physical mechanism responsible for the observed coherent radar echoes. The numerical simulations include the time evolution of electron density perturbations, which are respon sible for observed radar echoes, in the presence of dust layers in the mesosphere. Various dust parameters such as size, density and initial turbulence amplitudes are varied to estimate the electron density fluctuations. The results show that neutral air turbulence modulated dust particles known as fossil turbulence is responsible for the PMSE observed at four radar frequencies. This is the first time that results from radars operating at four frequencies are simultaneously employed to study the PMSE along with numerical simulations. The results discussed in the manuscript are of interest to Annals of Geophysicae community and I therefore recommend it for the publication. However, the authors have to implement the following suggestions before the manuscript becomes acceptable for publication. Specific Comments

1.The units of the radar intensity maps are different for figure 1 as compared to figure 2 and 3. Authors have to change the units such that all the figures are comparable.
We agree with the referee comments. Unfortunately, the MORRO radar is out of commission and the raw data is no longer available. While the units for the radar echoes at 8, 224 and 930 MHz can be changed to be consistent with the Figures 2 and 3, the 56 MHz data cannot be changed. Therefore, with all due respect, the authors prefer to keep Figure 1 in the current format. Please also note that the present paper provides the qualitative comparison of the data with the numerical simulations. Therefore, the suggested modification will not affect the conclusion of the paper. As mentioned by the referee this paper presents the first true common volume observations of PMSE source region with 4 radars.

2.How the neutral turbulence is related to the dust fluctuations is not clear. Authors have to discuss whether the spectrum of neutral turbulence and the dust particle fluctuations are same or not?
This has been addressed in an early work by Scales, 2004. Moreover, the neutral turbulence coupling with the dust cloud has been investigated in the previous work by Mahmoudian et al. (2017).

The following explanation has been added to the paper:

The effect of dust particles on density fluctuations in PMSE region was first investigated using a computational model by Lie-Svendsen et al. (2003). Their model treats plasma as fluid including arbitrary number of charged, and neutral particle species and dust/aerosol particles are modeled as particle in cell (PIC). Transport due to gravity, multipolar diffusion, and discrete charging model were also used (LieSvendsen et al., 2003). This model was used to explain the correlation and anticorrelation between electron and ion density fluctuations in the mesopause region. Scales (2004) developed a similar hybrid model including fluid plasma and particle in cell (PIC) dust with continuous dust charging process. This model is the first comprehensive model capable of studying the full physics of the PMSE. Continuous charging

model based on the Orbital-Motion-Limited (OML) approach has been used for the time varying charge on the dust particles. It should be noted that the difference between the continuous charging model and discrete charging model based on statistics is negligible in this circumstance. The summer mesopause temperature for both ions and electrons is taken to be $T_e = T_i =$ 150 K.

The collision of charged dust with neutrals is implemented by using a Langevin method (Winske and Rosenberg, 1998) and the dust-neutral collision frequency is denoted by $\nu_{dn}$. The initially uncharged dust is taken to have density given by

\begin{equation}
n_d(x) = n_{d0} \left( 1+ \frac{\delta n_{d0}}{ n_{d0}} \sin(2\pimxl)\right)
\end{equation}

Where $ n_{d0$ is the undisturbed density, $\deltan_{d0}$ is neutral dust irregularity amplitude, $m$ is the mode number, $l$ and is the system length of the model. In the current model, the plasma irregularities ultimately result from charging of the electrons onto this irregular dust density. The mechanism for the generation of the dust irregularities is presented in Mahmoudian et al. (2017).

D. Winske and M. Rosenberg, "Nonlinear development of the dust acoustic instability in a collisional dusty plasma," IEEE Trans. Plasma Sci., vol. 26, pp. 92–99, Feb. 1998.

Mahmoudian, A. , Mohebalhojeh, A. A.2 , Mazrae Farahani, M.2 and Scales, W. A., On the source of plasma density and electric field perturbations in PMSE and PMWE regions, Journal of the Earth and Space Physics, Vol. 42, No. 4, Winter 2017, PP. 63-71

Lie-Svendsen, Ø., Blix, T. A., Hoppe, U. P. and Thrane, E. V., 2003, Modeling the small-scale plasma response to the presence of heavy aerosol particles, Adv. Space Res., 31(9), 2045-2054

3.How the authors explain the absence of echoes at 930 MHz, based on fossil turbulence theory? The explanation given by the authors should be substantiated with further discussion. The whole discussion of the fossil turbulence is removed from the paper. This has also been requested by another referee.

4.The authors state that "Fluctuations in dusty plasma may also be generated by "fossil turbulence" when neutral air turbulence is absent". However they discuss the coupling of neutral turbulence and initial amplitude of irregularities within dust density. If fossil turbulence forms in the absence of neutral turbulence then how the coupling between them is justified.
We have revised the paper and removed the application of the fossil turbulence in the absence of neutral turbulence to the present study. It should be noted that the fluctuations in the dust and plasma density created through neutral turbulence may exist even when the initial turbulence is dissipated. This can be justified through reduced diffusion as a result of heavy dust particles. To keep the focus of the paper on neutral turbulence coupling and be consistent with the title of the paper the fossil turbulence is removed from the paper.

5.The discussion of the results is not very coherent and there are repetitions. Authors have to carefully go through the manuscript and try to avoid repetitions and firm up the discussion.
We agree with referee that the organization of the paper requires improvements. We have already incorporated in the revised version of the manuscript.

Minor Comments

line 145: evolution nor steady state → evolution of steady state
This has been corrected as requested by the referee.

line 191:low density → low dust density There is a scope for improving the English Grammar in the manuscript.
We have gone through the paper carefully to remove the grammatical mistakes.